# Prognostic Factors for Recurrence in Pituitary Adenomas: Recent Progress and Future Directions

**DOI:** 10.3390/diagnostics12040977

**Published:** 2022-04-13

**Authors:** Liang Lu, Xueyan Wan, Yu Xu, Juan Chen, Kai Shu, Ting Lei

**Affiliations:** Department of Neurosurgery, Tongji Hospital, Tongji Medical College, Huazhong University of Science and Technology, Wuhan 430030, China; tjluliang@tjh.tjmu.edu.cn (L.L.); xywan@tjh.tjmu.edu.cn (X.W.); deliaxu@gmail.com (Y.X.); jchen@tjh.tjmu.edu.cn (J.C.); kshu@tjh.tjmu.edu.cn (K.S.)

**Keywords:** pituitary adenoma, prognostic factors, recurrence, clinical factors, biomarkers, model

## Abstract

Pituitary adenomas (PAs) are benign lesions; nonetheless, some PAs exhibit aggressive behaviors, which lead to recurrence. The impact of pituitary dysfunction, invasion-related risks, and other complications considerably affect the quality of life of patients with recurrent PAs. Reliable prognostic factors are needed for recurrent PAs but require confirmation. This review summarizes research progress on two aspects—namely, the clinical and biological factors (biomarkers) for recurrent PAs. Postoperative residue, age, immunohistological subtypes, invasion, tumor size, hormone levels, and postoperative radiotherapy can predict the risk of recurrence in patients with PAs. Additionally, biomarkers such as Ki-67, p53, cadherin, pituitary tumor transforming gene, matrix metalloproteinase-9, epidermal growth factor receptor, fascin actin-bundling protein 1, cyclooxygenase-2, and some miRNAs and lncRNAs may be utilized as valuable tools for predicting PA recurrence. As no single marker can independently predict PA recurrence, we introduce an array of comprehensive models and grading methods, including multiple prognostic factors, to predict the prognosis of PAs, which have shown good effectiveness and would be beneficial for predicting PA recurrence.

## 1. Introduction

Pituitary adenomas (PAs) are benign lesions originating from different cell lineages of the anterior pituitary, representing 10–20% of all brain tumors [1]. The prevalence of PAs ranges from 80 to 100 per 100,000 person-years, while the incidence of clinically related PAs was 4.36 per 100,000 person-years, which is the second highest incidence rate among CBTRUS specific histologies [2,3]. PAs have been reported, by autopsy and radiological studies, to be more common than expected, with a total prevalence rate of 17% (range: 14–23%), many of which were discovered by accident [4]. Transnasal sphenoid sinus surgery represents the main treatment for PAs [5]. However, PAs invading the suprasellar or parasellar regions are difficult to completely remove, and 12–58% of patients with residual adenomas experience recurrence [1]. Even if the adenomas are completely resected, 10–20% will recur within 5–10 years [6]. The impact of pituitary dysfunction, invasion-related risks, and other complications considerably affects the quality of life of patients treated for recurrent PAs, and the standardized mortality rate will also increase [7]. Therefore, investigating predictive factors for recurrent PAs is of great value. This review summarizes research progress on two aspects—namely, the clinical and biological factors (biomarkers) for recurrent PAs. Moreover, comprehensive models and grading methods based on these prognostic factors are introduced to predict prognosis.

## 2. Clinical Factors

### 2.1. Postoperative Residue

Postoperative residue is the main factor in PA recurrence (Figure 1). The risk of recurrence of residual adenomas outside the sellar region was 3.7 times higher than the risk of recurrence of residual adenomas confined to the sellar region [8]. For patients with extrasellar or intrasellar residue, the recurrence rate 10 years after surgery was significantly higher than that 5 years after surgery (intrasellar: 58.3% vs. 15.4%; extrasellar: 76.9% vs. 51.4%) [8]. Therefore, more active treatment strategies should be adopted for patients with extrasellar residue.

A survey indicated that the recurrence rate in patients who underwent gross total resection (GTR) was 24%, and the regrowth rate in patients with surgical residue was 47% [9]. Additionally, the overall relapse rates at 5, 10, and 15 years were 25%, 43% and 61%, respectively [9]. A meta-analysis of more than 971 patients with PAs confirmed these findings and noted that there was a certain percentage of recurrence (12%) in the cohort with no residue even after a 10-year follow-up [1]. Previous reports have concurred that the absence of surgical residue is invariably associated with a lower risk of recurrence [10,11,12,13,14]. Indeed, 32% of patients with adenoma residue after surgery experienced recurrence within 2.2–6.3 years, while only 8% of patients without adenoma residue experienced recurrence within 5.0–6.5 years [9].

### 2.2. Age

Some evidence has shown that age may predict the long-term prognosis of PAs and that the possibility of recurrence may be higher in younger patients [15]. Watts reported that younger age was the main predictor of regrowth/recurrence in nonfunctional PAs (NFPAs) [14]. The regrowth rate was reduced by approximately 3% for every year of increase in patient age at presentation [14]. The regrowth rate remained 4.2 times higher in patients aged 41 years and younger compared with those aged >41 years [14]. Nevertheless, the question of whether age is a prognostic factor for PAs remains controversial, and the correlation between younger age and risk of relapse has been reported by some researchers [8,16,17,18,19], but not all [20,21,22,23].

### 2.3. Immunohistological Subtypes

The 2021 World Health Organization (WHO) Classification of Tumors of the Central Nervous System recognized special subtypes of PAs that exhibit more aggressive behaviors and have a higher risk of recurrence [24]. These special subtypes include immature PIT1-lineage adenomas, sparsely granulated somatotroph adenomas, Crooke’s cell adenomas, silent corticotroph adenomas, and lactotroph adenoma in men [25]. According to Brochier et al., plurihormonal adenomas, silent prolactin (PRL) adenomas, and silent growth hormone (GH)/adrenocorticotropic hormone (ACTH) adenomas relapse more frequently than gonadotropinomas and null cell adenomas [9]. Silent corticotroph adenomas have also been reported to show more aggressive behavior (including a higher recurrence rate) [26,27]. Among 814 patients with NFPA who underwent surgery, the recurrence rate and frequency of radiotherapy for silent ACTH and GH adenomas were significantly higher [28,29]. The size and invasiveness of silent ACTH adenomas was similar to that of silent gonadotropin adenomas (2.5 vs. 2.9 cm and 44% vs. 41%, respectively); however, more patients received radiotherapy (18% vs. 3%) and the recurrence rate was higher (36% vs. 10%) [28]. Fewer cystic adenomas (0.1% vs. 50%) and higher preoperative corticotropin levels (54 pairs of 28 pg/mL) were predictors of recurrence of silent ACTH adenomas [30]. Accurate classification can distinguish different subtypes of PAs according to their characteristics and prognosis, which would provide sound guidelines for therapy and patient follow-up in clinics [31,32,33,34].

### 2.4. Invasion

PAs are generally benign and slow-growing adenomas; nevertheless, the reported incidence of invasive PAs range from 35% to nearly 50% [1]. Invasive adenomas grow rapidly, have a high growth rate, can invade the cavernous sinus, sphenoid sinus, and sella turcica, and can even grow into the bone [35,36]. Bone invasion is significantly correlated with tumor regrowth in PAs, and more invasive adenomas are likely to have more extensive residue [37,38]. Several reports have indicated the interrelation between postoperative recurrence and invasive PAs [39,40,41,42,43,44,45]. In particular, 1 study reported that the recurrence rate for adenomas with and without cavernous sinus invasion (CSI) was 72.2% and 21.6%, respectively [45]. PAs easily invade the cavernous sinus because of the weak structure of the medial wall of the sinus [46], natural defects in the lateral wall of the pituitary fossa [46], and the lack of solid segmentation between the intracavernous internal carotid artery and PAs [47]. PAs located in the cavernous sinus are difficult to completely remove during surgery, resulting in a high recurrence rate. Chang et al. clearly indicated that PAs complicated by CSI showed a significant tendency to recur after surgery [23,48]. In addition, the degree of invasion could make a difference. Some researchers updated the Knosp grade by dividing grade 3 into grades 3A and 3B, with grade 3A referring to an extension into the upper cavernous sinus and grade 3B an extension into the lower cavernous sinus [49]. Grade 3B had a higher invasion rate than grade 3A (71% vs. 27%), and the total resection rate was negatively correlated with the Knosp grade (25% vs. 56%), suggesting that recurrence of adenomas in grades 3A and 3B may be different [49].

### 2.5. Tumor Size

The large size of adenomas could reduce the probability of total resection for adenomas, affecting the prognosis and recurrence rate. Hofstetter et al. examined the effect of adenoma size on the extent of adenoma resection and observed that PAs larger than 10 cm^3^ in size were more likely to have postoperative residue [50]. GTR was achieved in 90.2% of adenomas measuring <10 cm^3^ (vs. 40.0% of adenomas measuring >10 cm^3^) and was completed in 47.6% (vs. 9.1%). Specifically, GTR was performed in 85.3% (29/34) of patients with adenomas < 1 cm, 44.3% (31/70) with adenomas measuring 1–2 cm, 30.6% (19/62) with adenomas measuring 2–3 cm, 7% (4/57) with adenomas measuring 3–4 cm, and 15% (3/20) with adenomas >4 cm [50]. Other studies suggested that tumor size was associated with a higher risk of recurrence [38,42,51,52,53,54]. For instance, Lampropoulos reported that a large tumor diameter (≥25 mm) was a factor influencing the surgical outcome for both nonfunctioning and functioning adenomas [40]. In contrast, Ferreira et al. expressed a different opinion in their study [55]. They studied 117 patients with NFPAs and showed that the maximum diameter of adenomas was not associated with the recurrence rate [55]. Such different results may be attributable to differences in adenoma subtypes and different study designs.

### 2.6. Hormone Levels

Functional PAs are often accompanied by abnormal hormone levels, and different hormone levels often reflect the characteristics of PAs, which may have different effects on the surgical outcome. Previous studies, particularly on ACTH adenomas, suggested that preoperative or postoperative hormone levels could predict postoperative recurrence [16,51,56,57,58]. Morning serum cortisol levels on postoperative day 1 was reported to predict long-term remission in patients with ACTH adenomas by Clayton et al. [56]. A survey indicated that recovery from transient postsurgical adrenal insufficiency (2–34 months) predicted a low recurrence rate (13%), while the absence of a diurnal cortisol secretion rhythm predicted a significantly higher recurrence rate (50–65%) [58]. Using different machine learning algorithms, Liu et al. showed that young age, postoperative ACTH levels, and postoperative serum cortisol levels were important predictors of recurrence [59]. Furthermore, a retrospective study involving 41 patients with ACTH adenomas revealed that a higher preoperative ACTH level was a predictor of recurrence [16]. A thorough meta-analysis suggested that low cortisol levels immediately after surgery appeared to be a positive prognostic indicator of long-term remission according to the majority of reports [51]. 

### 2.7. Postoperative Radiotherapy

Postoperative radiotherapy has a certain effect on recurrence. According to Ozgen et al., radiotherapy after resection of PRL adenomas could prolong the time of progression or recurrence and significantly reduced the recurrence rate from 22% to 8% [60]. Chang et al. reported the long-term follow-up results of 663 patients with NFPAs after surgery and adjuvant radiotherapy [23]. The recurrence rate was 9.7%, while the recurrence rates at 5, 10, and 15 years were 93%, 87%, and 81%, respectively [23]. Brochier et al. confirmed that immediate postoperative radiotherapy was independently associated with a much lower risk of regrowth/recurrence [9]. Furthermore, the recurrence rate in patients who did not receive radiotherapy gradually increased during the follow-up period, reaching 72% after 15 years; in contrast, the recurrence rate in patients who received radiotherapy stabilized at 9% after 10 years [9]. At long-term follow-up, radiotherapy could reduce the recurrence rate in patients who underwent subtotal resection; it could not only affect the recurrence rate but also increase the risk of death in patients who underwent GTR [61]. Radiotherapy could inhibit the growth of adenomas, reduce the recurrence rate of postoperative adenomas, and improve endocrine function in patients with PAs [10,11,12,21,62,63,64,65,66]. However, the onset time of radiotherapy for PAs was relatively slow, and it often caused complications such as hypothalamic dysfunction, visual impairment, and hypopituitarism [61]. Therefore, radiotherapy was not recommended as first choice for PA treatment, but as an adjuvant treatment for patients with abnormal hormone levels or postoperative adenoma residue [66].

## 3. Biomarkers

In addition to clinical factors, some researchers have recently attempted to identify biomarkers related to PA recurrence in order to guide clinical diagnosis and therapy. These biomarkers come from a wide range of biological fields, including proliferation markers, growth factor receptors, cell adhesion-related factors, miRNAs, lncRNAs, and other specific molecules (Figure 2). Although the exploration of various aspects has shown potential predictive value, their effectiveness in predicting PA recurrence still requires verification. Thus far, no single marker has been confirmed to reliably predict the recurrence behavior of PAs.

### 3.1. Ki-67 and p53

Ki-67 is a proliferation marker widely expressed in the different phases of the cell cycle and is the most widely studied protein in MIB-1 immunoassay. The Ki-67 labeling index for PAs is mostly 1–2% [67]. Quantification of the Ki-67 labeling index can distinguish pituitary carcinomas (11.9 ± 3.4% on average) and other adenomas (1.4 ± 0.15% on average) [68]. Therefore, PAs with a Ki-67 labeling index > 10% should be routinely classified as atypical PAs [69,70]. Almeida reported that Ki-67 > 5% was associated with a higher probability of PA recurrence [41].

Similarly, p53 nuclear protein, a common tumor inhibitor, is not mutated in most PAs but is expressed in 100% of pituitary carcinomas [69]. In 2004, the WHO utilized p53, Ki-67, and mitotic activity to distinguish a class of highly malignant PAs from ordinary PAs (namely, atypical PAs) [71]. Nonetheless, subsequent studies revealed that this concept could not always accurately distinguish PAs with poor prognosis and predict which PAs might be difficult to deal with [72]. Therefore, in 2017, the WHO abandoned this classification in the new classification system for PAs [24]. Whether Ki-67 is an efficient prognostic factor for PAs remains controversial. In a review involving 28 studies on Ki-67, 18 studies reported high Ki-67 expression in recurrent adenomas, while the other 10 studies showed no correlation [73]. In contrast, Oliveira et al. reported no obvious correlation between p53 and PA recurrence. In particular, the positivity rate of p53 in 148 patients with PAs was only 1.3%, implying that p53 was insufficient to be used as a routine marker of PA recurrence tendency [74].

### 3.2. Cadherin

The loss of cell–cell adhesion mediated by cadherin often changes the behavioral characteristics of PAs [75]. A range of molecular changes associated with epithelial–mesenchymal transformation may affect PA recurrence [76,77]. E-cadherin, β-catenin, and H-cadherin are decreased in PAs, while N-cadherin and γ-catenin are increased [76,78,79,80,81,82,83,84,85,86]. β-catenin may induce PA invasion and proliferation via regulation of ERK/MAPK, and matrix metalloproteinase (MMP)-2/MMP-9 [84]. Another study revealed that β-catenin expression was significantly associated with NFPA recurrence [85]. Fibroblast growth factor receptor-4 was associated with decreased expression of membrane N-cadherin and induced PA invasion in living animal models [86]. In addition, some studies showed that the overexpression of EpCAM (epithelial cell adhesion molecule) and Trop2 (adenoma-associated calcium signal transduction) was significantly associated with the invasiveness and proliferation of PAs and that they could be used as predictors of PA recurrence [87].

### 3.3. Pituitary Tumor Transforming Gene (PTTG)

PTTG, a cell-cycle regulator and transcriptional activator, was also associated with PA recurrence [88]. Vassallo pointed out that PTTG expression was positively correlated with the Ki-67 index of PAs and that the positive rate of PTTG was positively correlated with the suprasellar range and volume of PAs [77]. Other studies reported the association between nuclear PTTG expression and tumor recurrence [89,90,91,92], that suggested PTTG expression could be used as a marker for the enhancement of PA proliferation. A study also showed that the antiproliferative effect of somatostatin analogues in vivo was achieved via regulation of the cell cycle [92].

### 3.4. MMP-9

MMPs are a family of proteinases that regulate the extracellular matrix and are considered to be key molecules that promote PA invasion [93,94,95,96]. The expression of MMP-9, the most studied MMP in PAs, was reported to be significantly increased in some invasive and recurrent PAs, as well as in the majority of pituitary carcinomas [94,96,97,98]. In the study conducted by Wang et al., patients with ACTH adenomas who had higher MMP-9 levels showed higher recurrence rates and shorter recurrence intervals, suggesting that MMP-9 could be used as a valuable tool for predicting the recurrence of ACTH adenomas [96].

### 3.5. miRNAs and lncRNAs

Recent studies on miRNAs and lncRNAs have afforded new insight into PA recurrence [99,100,101,102,103]. A study conducted by Butz with patients with gonadotropin adenomas revealed that a decrease in plasma miR-143-3p was associated with an increase in the total resection rate, leading to a significant increase in progression-free survival [99]. Patients with downregulated miR-193a-3pA experienced a higher risk of postoperative residue and recurrence [100]. However, there was a low presence of differentially expressed miRNAs in PAs, thereby reducing its role as a biomarker. RPSAP52, an antisense lncRNA of the HMGA2 gene, was highly upregulated in gonadotropin and PRL PAs, which could encourage cell growth by promoting G1Mel S transition in the cell cycle [101]. Cheng et al. reported a prognostic signature involving three lncRNAs (LOC101927765, RP11-23N2.4 and RP4-533D7.4) with high prediction accuracy for tumor recurrence [102].

### 3.6. Others

Other specific molecules can also be used as markers to predict the tendency of PA recurrence. Overexpression of cyclooxygenase-2 (COX-2), which is an enzyme that catalyzes prostaglandin formation from arachidonic acid, has been shown to increase cell proliferation and angiogenesis [104]. Recent studies have demonstrated COX-2 overexpression in various human tumors and have shown its correlation with higher tumor stage, larger tumor size, and risk of recurrence in PAs [105]. Akbari reported that COX-2 had a significant expression level in NFPAs, and the COX-2 expression level was significantly increased in macroadenomas and invasive adenomas [106].

The epidermal growth factor receptor (EGFR) is a transmembrane tyrosine kinase receptor that signals key cellular functions including proliferation and differentiation [107,108]. EGFR expression has been shown to be positively correlated with ACTH and cortisol levels and adenoma recurrence [109]. Moreover, EGFR overexpression, caused by the USP48 and BRAF genes, could subsequently enhance cell proliferation and tumor growth in PAs [110]. According to some studies, EGFR is associated with PRL secretion, tumor size and invasion, and risk of recurrence in human PRL adenomas and in animal models [111,112].

Fascin actin-bundling protein 1 (FSCN1) was implicated in an increased risk of metastasis in various human cancers [113]. In some studies, the positive rate of FSCN1 was significantly higher in the invasive group than in the non-invasive group (54.6% vs. 18.3%), and the recurrence rate was 19.7% for the high-expression group and 5.7% for the low-expression group [114,115].

## 4. Comprehensive Models

PAs are benign; nonetheless, their classification and characteristics are complex. According to a large number of studies, no single marker could independently predict the recurrence tendency of PAs [39]. Therefore, the use of comprehensive models and grading methods, including multiple prognostic factors, to predict the prognosis of PAs has become a trend and can aid in accurately judging the curative effect and long-term prognosis in clinics (Table 1).

In a study involving 501 surgical patients, Pappy used three important parameters to establish a predictive model—namely, adenoma diameter, CSI, and Ki-67 [116]. The results indicated that this model could predict the development of long-term events. Specifically, with CSI, diameter ≥ 2.9 cm, and Ki-67 > 3 as the cutoff value, the specificity of predicting persistent hypersecretion syndrome and residual adenoma was 98.7% (odds ratio, 8.6; confidence interval, 3.0–24.7), which showed an excellent predictive effect. As an early risk stratification system, this model was beneficial for individual monitoring and treatment of patients with PAs. Furthermore, these three parameters were the most closely related factors; nevertheless, each factor might have limitations and it was impossible to independently predict the pathogenesis of adenomas. Therefore, multiple factors were used to evaluate them from a more comprehensive point of view, which could be closer to the real efficacy and results. 

Wang et al. applied an advanced gene sequencing technique to establish a prognostic model for predicting early PA recurrence [117]. Using mRNA-Seq data and clinical data on early postoperative PA recurrence, patients were randomly divided into training and verification cohorts. A seven-gene predictive model was established. Of the seven significant genes, six (SPRY3, ZNF343, GZF1, C15orf61, SLC24A4, HOXB5) were highly associated with early PA recurrence. The area under the curve (AUC) was 0.857 for the training group, and 0.936 for the verification group. Patients with low-risk scores were significantly less likely to have early postoperative recurrence than those with high-risk scores, showing a good predictive power. Thus, this contributes to the classification of PAs and the implementation of appropriate treatment and follow-up strategies for patients with PAs.

Trouillas et al. selected a comprehensive combination of invasion and proliferation to predict postoperative PA recurrence [118,119]. In an 8-year, retrospective, multicenter, cohort study that included 410 patients [118], they observed that the 2 graded components of invasion and proliferation were independent, with invasiveness being the main prognostic factor for predicting progression-free status. Furthermore, they showed that a comprehensive evaluation model of the two graded components could predict relapse/progression-free status (AUC, 81.4%) in patients with PAs. Specifically, PAs were divided into 4 categories according to invasion (non-invasive, “1”; invasive, “2”) and proliferation (non-proliferative, “a”; proliferative, “b”). The evaluation criterion for invasion was Knosp grades 3–4, while the evaluation criterion for proliferation was the presence of at least 2 of the following 3 proliferation markers: Ki-67 ≥3%, p53 staining > 10 strong positive nuclei/10 times high magnification visual field (HPF), and mitosis > 2/10 HPF. The results indicated that invasion and proliferation had a synergistic effect on refractory adenomas and that the effect of invasion on progression-free survival was greater than that of proliferation. Specifically, compared with grade 1a PAs, the risk of relative persistent disease was 8.0 for grade 2a PAs and 3.1 for grade 1b PAs. Grade 2b PAs had a 25-fold higher risk of persistent disease and a 12-fold higher risk of adenoma progression than grade 1a PAs. In addition to the above-described 8-year retrospective cohort study, one study also validated this classification in a prospective single-center cohort comprising 374 postoperative patients followed up for 3.5 years [119]. Multiple cohort studies confirmed the prognostic value of this classification in other study groups [120,121,122].

A nomogram is a model used to estimate prognosis in oncology by summing the scores for each risk factor, and graphically demonstrating the combined effect of each factor [123]. Lyu et al. constructed a nomogram to predict the postoperative recurrence of NFPAs [124]. Age, CSI, tumor size, sphenoid sinus invasion, and surgical extension were included in the nomogram (AUC, 0.953). Moreover, 172 patients with large or giant PAs were involved in a retrospective study that aimed to develop a prognostic nomogram based on the extent of resection, body mass index, Ki-67, Knosp classification, and smoking [125]. The AUCs for 1-, 2-, and 3-year survival were 0.889, 0.885, and 0.832, respectively [125]. This form of comprehensive model made efficient use of effective prognostic factors; however, both studies had a single-center study design and lacked internal and external verification cohorts, which influenced the generalizability of the model [126]. In addition, prospective models for other subtypes of PAs would be beneficial for future reference, and it is also hoped that more valuable prognostic factors would be included in the models.

**Table 1 diagnostics-12-00977-t001:** Summary of recent studies on comprehensive model for predicting the recurrence of pituitary adenomas.

References	Content of the Models	Form	Sample Size	Prediction Performance
Pappy A. L. et al., 2019 [116]	Model 1 (CSI, diameter ≥ 2.9 cm and ki-67 > 3%)	Prognostic model	Training (n = 501)	98.7% specificity (OR 8.6; CI 3.0–24.7)
	Model 2 (ki-67 > 3% and CSI)			93.1% specificity (OR 3.3; CI 1.8–6.0)
	Model 3 (ki-67 > 3%, mitoses and p53, former “atypical” adenoma)			96.0% specificity (OR 2.3; CI 1.0–5.0)
Wang X. et al., 2019 [117]	SPRY3, ZNF343, GZF1, C15orf61, SLC24A4, HOXB5, SLC9A3R2	Prognostic model	Training (n = 57) Validation (n = 50)	- Training (AUC 0.857) - Validation (AUC 0.936) - All (AUC 0.848)
Trouillas J. et al., 2013 [118]	- Grade 1a: non-invasive tumour - Grade 1b: non-invasive and proliferative tumour - Grade 2a: invasive tumour - Grade 2b: invasive and proliferative tumour - Grade 3: metastatic tumour (cerebrospinal or systemic metastases)	Grading classification	Training (n = 410)	Invasion + proliferation (AUC = 81.4%);
				Invasion + Ki-67 ≥ 3% (AUC = 81.4%)
				Proliferation without invasion (AUC = 0.713);
				Ki-67 ≥ 3% without invasion (AUC = 0.711)
Wen L. et al., 2021 [124]	- Age (HR = 0.50), - Tumor size (HR = 11.06), - CSI (HR = 7.53), - SSI (HR = 13.14), - GTR (HR = 0.12)	Nomogram	Training (n = 145)	AUC = 0.953
				A well-fitted calibration curve
Chen Y. et al., 2021 [125]	- Smoking history (HR = 3.10), - BMI ≥ 25 kg/m^2^ (HR = 2.00), - Knosp grade 4 (HR = 4.09), - partial resection (HR = 3.72), - Ki-67 ≥ 3% (HR = 4.64)	Nomogram	Training (n = 172)	AUC for 1-, 2-, and 3-year survival (0.889, 0.885 and 0.832, respectively)
				Well-fitted calibration curves

CSI: cavernous sinus invasion; HR: hazard ratio; OR: odds ratio; CI: confidence interval; AUC: area under the curve; PAs: pituitary adenomas; SSI: sphenoid sinus invasion; GTR: gross-total resection; BMI: body mass index.

## 5. Conclusions

PA recurrence is an important source of refractory PAs, which deserves great clinical attention. If patients at high risk for PA recurrence could be identified early, this would help clinicians to adjust their treatment and improve the long-term cure rate. This review summarized two aspects of related factors for PA recurrence—namely, clinical factors and biological markers—and introduced several different factors that showed predictive ability in previous studies. Unfortunately, no single marker could reliably predict PA recurrence. In addition, multi-factor models have been used to predict the recurrence behavior of PAs and have achieved good results, which is an interesting trend. Although PAs are benign, their classification and characteristics are complex and changeable. In a sense, this comprehensive model and grading method is an excellent way to help clinics correctly understand and judge prognosis and curative effect, and it is also the direction of progress that can be popularized in the future.

## Figures and Tables

**Figure 1 diagnostics-12-00977-f001:**
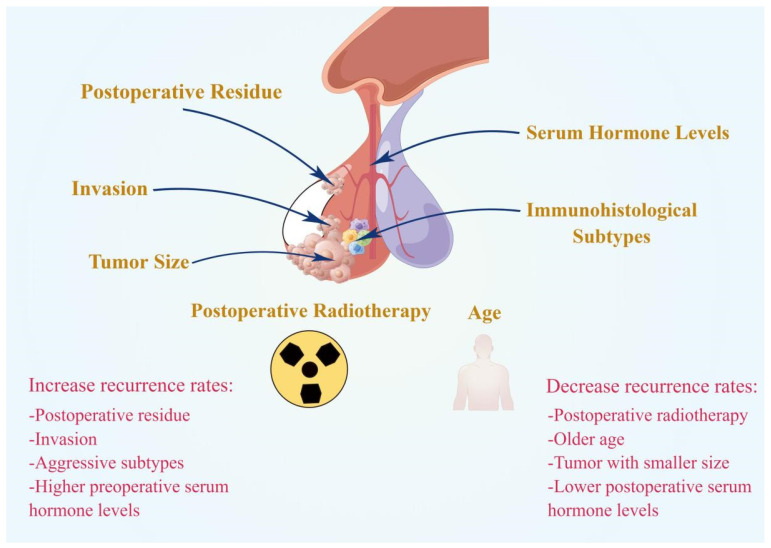
Summary of clinical prognostic factors for PA recurrence. Postoperative residue, invasion, aggressive subtypes, and higher preoperative serum hormone levels could promote the recurrence of PAs. Whereas postoperative radiotherapy, older age, smaller tumor size, and lower postoperative serum hormone levels could inhibit the recurrence of PAs.

**Figure 2 diagnostics-12-00977-f002:**
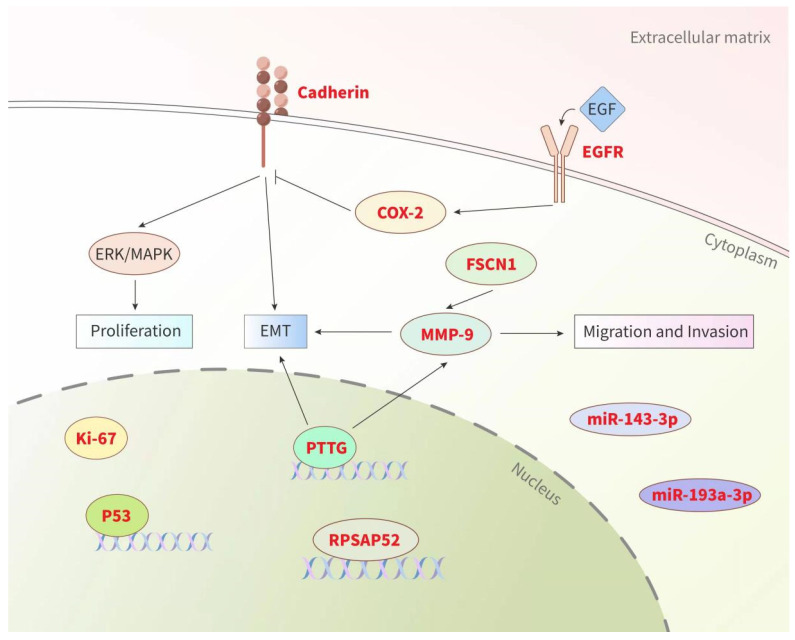
Summary of biological factors (biomarkers) for PA recurrence. Prognostic biomarkers (highlighted in red) from various sources could be used as valuable tools for predicting the recurrence of PAs (see text for details). Pituitary tumor transforming gene (PTTG), matrix metalloproteinase (MMP), cyclooxygenase-2 (COX-2), epidermal growth factor receptor (EGFR), fascin actin-bundling protein 1 (FSCN1), epithelial mesenchymal transformation (EMT), extracellular signal-regulated kinase (ERK), mitogen-activated protein kinase (MAPK).

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
