# Peer review of "Prognostic Factors for Recurrence in Pituitary Adenomas: Recent Progress and Future Directions"

_diagnostics, 2022, doi:10.3390/diagnostics12040977_

Round 1
Reviewer 1 Report
The ms by Lu et al is a comprehensively written focused review on prognostic factors for recurrence in pituitary adenomas. There is not a lot such summaries in the literature.
I would suggest the following aspects to reconsider:
- While speaking about the epidemiology it would be better to introduce the newest statistical reports (Ostrom QT, Cioffi G, Waite K, Kruchko C, Barnholtz-Sloan JS. CBTRUS Statistical Report: Primary Brain and Other Central Nervous System Tumors Diagnosed in the United States in 2014-2018. Neuro Oncol. 2021 Oct 5;23(12 Suppl 2):iii1-iii105. doi: 10.1093/neuonc/noab200).
- Ref. [24] is the 2007 class, referring in the text to 2017 class. Both are outdated = Louis DN, Perry A, Wesseling P, Brat DJ, Cree IA, Figarella-Branger D, Hawkins C, Ng HK, Pfister SM, Reifenberger G, Soffietti R, von Deimling A, Ellison DW. The 2021 WHO Classification of Tumors of the Central Nervous System: a summary. Neuro Oncol. 2021 Aug 2;23(8):1231-1251. doi: 10.1093/neuonc/noab106.
- Figure 1 - visually nice picture but lacking a meaning. the arrows from 3 factors indicate adenoma, the rest of the lines has no arrows at all. Some reedition should be done to better visualize the clinical prognostic factors for the recurrence (small tables on both sides?, small arrows on the lines indicating what increases or decreases the possibility of recurrence?) = no special suggestions, as the authors wish, as long as the picture represents the idea.
- Table 1 - the columns should be reorganized - reference as the last, first name/content of model, form, sample size, prediction performance.
Reviewer 2 Report
Lu et al provide a narrative review on factors that are associated with recurrence in pituitary adenomas.
The paper is of interest and, even if it is limited by its design, it provides a balanced overview of the study object.
English revision would correct some minor issues.
It would be nice to improve the quality of the figures and to improve the visualization of the table.
Otherwise, I believe the paper would provide a useful review for the reader who is looking for a good introduction to the issue of recurrence in pituitary adenomas.
